# Assessment of EULAR/ACR-2019, SLICC-2012 and ACR-1997 Classification Criteria in SLE with Longstanding Disease

**DOI:** 10.3390/jcm10112377

**Published:** 2021-05-28

**Authors:** Berta Magallares, David Lobo-Prat, Ivan Castellví, Patricia Moya, Ignasi Gich, Laura Martinez-Martinez, Hye Park, Ana Milena Millán, Ana Laiz, César Díaz-Torné, Susana Fernandez, Hèctor Corominas

**Affiliations:** 1Department of Rheumatology, Hospital de la Santa Creu i Sant Pau, 08025 Barcelona, Spain; bmagallares@santpau.cat (B.M.); Dlobo@santpau.cat (D.L.-P.); icastellvi@santpau.cat (I.C.); pmoyaa@santpau.cat (P.M.); HyeSang.Park@sanitatintegral.org (H.P.); amillana@santpau.cat (A.M.M.); alaiz@santpau.cat (A.L.); cdiazt@santpau.cat (C.D.-T.); sfernandez@santpau.cat (S.F.); 2Sant Pau Biomedical Research Institute (IIB Sant Pau), 08025 Barcelona, Spain; igich@santpau.cat; 3Department of Immunology, Universitat Autònoma de Barcelona (UAB), 08193 Barcelona, Spain; 4CIBER Epidemiología y Salud Pública (CIBERESP), 28029 Madrid, Spain; 5Department of Clinical Epidemiology and Public Health, Hospital de la Santa Creu i Sant Pau, 08041 Barcelona, Spain; 6Department of Immunology, Hospital de la Santa Creu i Sant Pau, 08025 Barcelona, Spain; lmartinezm@santpau.cat

**Keywords:** SLE, classification criteria, EULAR/ACR-2019, SLICC-2012, ACR-1997, longstanding lupus

## Abstract

Background: Different classification criteria for systemic lupus erythematosus (SLE) have been launched over the years. Our aim was to evaluate the performance of the EULAR/ACR-2019, SLICC-2012 and ACR-1997 classification criteria in a cohort of SLE patients with longstanding disease. Methods: Descriptive observational study in 79 patients with established and longstanding SLE. The three classification criteria sets were applied to those patients. Results: Of the 79 patients, 70 were women (88.6%), with a mean age of 51.8 ± 14 years and a mean disease duration of 15.2 ± 11.5 years. The sensitivity of the different criteria were: 51.9%, 87.3% and 86.1% for ACR-1997, SLICC-2012 and EULAR/ACR-2019, respectively. In total, 68 out of 79 patients (53.7%) met all three classification criteria; 11.4% did not meet any classification criteria and were characterized by low SLEDAI (0.6 ± 0.9), low SLICC/ACR Damage Index (0.88 ± 0.56) and fulfilling only skin domains, antiphospholipid antibodies or hypocomplementemia. To fulfill EULAR/ACR-2019 criteria was associated with low complement levels (*p* < 0.04), high anti-dsDNA levels (*p* < 0.001), presence of lupus nephritis III-IV (*p* < 0.05) and arthritis (*p* < 0.001). Conclusion: The EULAR/ACR-2019 classification criteria showed high sensitivity, similar to SLICC-2012, in SLE patients with longstanding disease. Patients with serological, articular or renal involvement are more likely to fulfill SLICC-2012 or EULAR/ACR-2019 criteria.

## 1. Introduction

Systemic lupus erythematosus (SLE) is a systemic autoimmune disease with a wide range of clinical manifestations occasionally leading to life-threatening organic failure [1]. The diagnosis of SLE may be challenging as several other conditions can mimic SLE and there are no specific findings to set up the diagnosis [2,3]. SLE is based on the sum of signs, symptoms, serological parameters, radiological features and histologic and pathological findings [4]. 

Classification criteria are essential to identify well-defined, relatively homogeneous groups of patients; they are primarily designed to be used in clinical research [5]. Classification criteria do not mean diagnostic criteria, but they are used frequently to detect patients with clinical symptoms and laboratory features of the disease [6]. Although they can provide diagnostic aids, the classification criteria are characterized by high specificity and usually lower sensitivity; therefore, patients with very recent onset of the disease or with less common manifestations may be missed [7]. Longstanding SLE cohorts should mostly meet these criteria, given the fact that they are patients with established disease and the risk of them presenting other systemic autoimmune diseases is slightly lower. 

In 1971, the American College of Rheumatology (ACR) published the first preliminary criteria for SLE. These criteria were later updated in 1982 [8] and in 1997 [9]. The 1997 ACR criteria maintained the same structure, with 11 criteria, both clinical and immunological, four of which had to be present to identify patients with SLE [8,9]. Some clinical characteristics were over-represented, such as skin manifestations covering four of the 11 criteria, while immunological criteria were represented by only two criteria: positivity for Antinuclear antibodies (ANA)s on the one hand, and the presence of anti-double-stranded DNA (anti-dsDNA), anti-Sm or antiphospholipid antibodies on the other [9]. Consistently, ACR-1997 criteria have a sensitivity of 85% and specificity of 98% for classifying patients as having SLE [5,10]. 

Taking into account the limitations of the ACR-1997 criteria, the Systemic Lupus International Collaborating Clinics (SLICC) group published a new classification criteria proposal in 2012. The SLICC criteria were launched with 11 clinical and six immunological items. These criteria better determined each criterion and included some of the characteristic mucocutaneous manifestations and neuropsychiatric symptoms in comparison with ACR-1997 classification criteria. The most important advance in the SLICC-2012 criteria was the inclusion of specific histological findings of lupus nephritis, which along with immunological findings, are nowadays a sufficient condition to meet the classification criteria [11]. The SLICC-2012 criteria reached a sensitivity of 95% with a specificity of 95.5% [10].

Recently, a major effort to develop new criteria was led by the European League Against Rheumatism (EULAR) and ACR, and in 2019, new classification criteria came to light [12]. There are two main particularities among this new set of criteria in comparison with previous proposals: the required ANA positivity as an entry criterion and the categorization of the other criteria in different domains weighted from 2 to 10, where only the highest item of each domain is counted [11]. At least one clinical criterion is required and a total score of 10 points is necessary to be classified as having SLE [12]. Another key feature is that items are only counted towards SLE if there is no other more likely explanation [11,12]. One of the goals of these new criteria was to maintain high specificity, such as the ACR-1997 criteria, and high sensitivity, similar to the SLICC-2012 criteria. The validation data concluded that this objective was met, with a sensitivity of 96% and a specificity of 93% [13]. Since their publication, the precision of the new criteria has been evaluated in different SLE cohorts, such as early [5,14] and pediatric SLE [15,16]. Nevertheless, longstanding SLE patients represent the majority of patients in clinical practice, with various disease severities and usually exposed to several treatments. As patients with longstanding SLE are also potential candidates for clinical trials, it is crucial to evaluate how classification criteria perform in this subset of patients. Furthermore, these patients have an established disease and time is a key factor ruling out other diseases that could mimic SLE in their onset. Efforts have been made to assess the EULAR/ACR-2019 criteria in real clinical practice [17,18], but their performance and application in long-term SLE have not yet been well established.

In the present study, we aimed to assess the performance and characteristics of the EULAR/ACR-2019 in comparison with SLICC-2012 and ACR-1997 classification criteria in a cohort of SLE patients with longstanding disease.

## 2. Materials and Methods

The study was carried out in the Systemic Autoimmune Disease Outpatient Clinic at the Rheumatology Department of Hospital Universitari de la Santa Creu i Sant Pau. Our center is the only tertiary referral hospital in the district of AIS Dreta in Barcelona with around 308,268 inhabitants. All high complexity and severe diseases are referred to our center through the Universal Health Coverage of the Public Health System.

The inclusion criteria were patients older than 18 years with a confirmed diagnosis of SLE based on the clinical expertise of the attending rheumatologist, with a follow-up of at least 12 months. The exclusion criteria were patients who had lost follow-ups or for any reason did not provide necessary clinical or laboratory information to evaluate the classification criteria’s fulfillment. Between June 2020 and August 2020, 84 patients were screened during routine clinical follow-ups. Finally, 79 consecutive patients were included. 

Demographic, clinical and laboratory data were gathered. Disease activity was measured using the Systemic Lupus Erythematosus Disease Activity Index (SLEDAI) and accumulated damage was assessed using the SLICC/ACR Damage Index (SLICC/ACR DI). Each of the three sets of criteria was applied at the last follow-up after reviewing past medical records and SLE damage.

Anti-nuclear antibodies were determined by indirect immunofluorescent (IIF) assay with Hep2 cells (INOVA diagnostics). Screening dilution was over 1:80. The anti-dsDNA and anti-Sm and antibodies were tested using chemiluminescence tests (INOVA diagnostics). The cut-off of positivity was 35 Units/mL for anti-dsDNA and 20 Units/mL for anti-Sm. The quantitative determination of antiphospholipid IgG/IgM was performed using Orgentec Diagnostika GmbH (Palex) kits with the ELISA method, following the recommendations of the International Society on Thrombosis and Haemostasis. To ensure a homogeneous method of immunological examination, all tests were performed in the same laboratory and repeated before inclusion. All patients had blood tests checked at least every 6–12 months except for patients with more active disease who required more frequent examination.

The characteristics of the cohort were presented as absolute and relative frequencies [n (%)] for categorical variables, and using mean and standard deviation (mean ± SD) for quantitative variables. A study of association between criteria fulfillment and clinical and immunological data was performed using contingency tables and chi-square; to calculate concordance between the criteria, the kappa coefficient was used. Statistical significance was assumed in *p* values < 0.05. IBM-SPSS (V26.1) software was used for data analysis. 

## 3. Results

A total of 79 patients were included. Seventy patients (88.6%) were women. The mean age of our cohort was 51.8 ± 14 years and they had a mean disease duration of 15.2 ± 11.5 years. All patients had a positive ANA test. The clinical and demographic characteristics of the cohort are detailed in Table 1. 

Out of 79 patients, 40 (51.9%) met all three classification criteria. Figure 1 shows the proportion of patients who met the sets of different classification criteria. In our cohort, no patients met the EULAR/ACR-2019 and ACR-1997 but not the SLICC-2012, nor did we find any patients who met only the ACR-1997 criteria. Nine patients (11.4%) did not meet any SLE classification criteria. Intriguingly, those patients were characterized by meeting only skin domains (alopecia or oral ulcers), antiphospholipid antibodies or hypocomplementemia domains, and presented low SLEDAI scores (0.6 ± 0.9). In addition, they presented significantly lower SLICC/ACR Damage Index scores compared to those who met all three sets of criteria (0.88 ± 0.56 vs. 4.02 ± 0.56; *p* < 0.025).

Seven patients only fulfilled one of the three classification criteria: three of them only met the EULAR/ACR-2019 criteria and were characterized by having hypocomplementemia or arthritis with positive anti-dsDNA and ANA. The four patients that exclusively met the SLICC-2012 criteria did not have a predilection for any domain. The cohort of patients who met the EULAR/ACR-2019 criteria had high scores, almost doubling the cut-off point for classifying in comparison with the ACR 1997 and SLICC 2012 sets of criteria. Interestingly, patients with higher scores (≥20) for the EULAR/ACR-2019 classification criteria were characterized by higher scores on the SLICC/ACR Damage Index (4.41 ± 1.13) compared to those with lower EULAR/ACR-2019 scores (2.54 ± 1.01 SD) (*p* < 0.02).

The percentages of patients meeting the different classification criteria are summarized in Table 2.

Regarding the EULAR/ACR-2019 criteria, statistically significant associations were found between meeting the EULAR/ACR-2019 classification criteria and the presence of low C3 and C4 (*p* = 0.03), anti-dsDNA (*p* < 0.01), lupus nephritis III-IV (*p* < 0.05) and arthritis (*p* < 0.01). Only one patient with C3 and C4 hypocomplementemia did not meet any of those criteria. All patients with positive anti-dsDNA met the criteria and 73.5% of patients that met the EULAR/ACR-2019 criteria tested positive for anti-dsDNA. All patients with lupus nephritis III–IV and arthritis were classified as having SLE, and 55.9% and 17.6% of patients classified as having SLE by these criteria had arthritis or lupus nephritis III–IV, respectively. Moreover, all patients with proteinuria > 0.5 g/24 h also met the EULAR/ACR-2019 criteria.

In the SLICC-2012 evaluation, there were significant associations between meeting these criteria and the presence of arthritis (*p* < 0.01), renal involvement (*p* < 0.01), leukopenia/lymphopenia (*p* < 0.01), anti-dsDNA (*p* = 0.02) and hypocomplementemia (*p* = 0.02). Of the patients with arthritis and/or renal involvement/leukopenia/lymphopenia, 97.2% and 100% were classified as having SLE, respectively. Likewise, 94% of the patients with elevated anti-dsDNA and 93% with decreased complement also met the SLICC-2012 criteria. 

Fulfillment of ACR-1997 was associated with malar rash (*p* < 0.01), discoid rash (*p* = 0.05), oral ulcers (*p* = 0.03) and presence of photosensitivity (*p* < 0.01), as well as arthritis (*p* < 0.01), serositis (*p* = 0.02) and renal (*p* = 0.05) and hematologic (*p* = 0.05) involvement. The proportions of patients in our cohort with malar rash, discoid lupus, oral ulcers or photosensitivity who met the ACR-1997 criteria were 93%, 85.7%, 67.8% and 84.6%, respectively. Of the patients with arthritis, 73.6% were classified as having SLE, as well as 100%, 68% and 72.4% of those with serositis, renal and hematologic involvement, respectively. Table 3 summarizes the different involvement associations with all sets of classification criteria.

Finally, the Kappa agreement coefficient between the three sets of classification criteria found the best concordance between the EULAR/ACR-2019 and SLICC-2012 classification criteria (K 0.61) in comparison with the agreement between the EULAR/ACR -2019 and ACR-1997 (K 0.27) and between the SLICC-2012 and ACR-1997 (K 0.30).

## 4. Discussion

We found significant differences among the three sets of SLE classification criteria and the patients’ characteristics according to the achieved criteria. Only 51.9% of our patients met all three classification criteria, which is a lower proportion than that described in other cohorts [5,13,14]. This percentage is even lower than expected considering that those were patients with long-standing disease [19]. It is known that the sensitivity of each classification criteria differs among them. The EULAR/ACR-2019 classification criteria show high sensitivity in SLE patients with longstanding disease, similar to the SLICC-2012 criteria, both of which are much higher than the ACR-1997, as has been described previously in other cohorts [5,13,14]. This might be explained by the increased weight that a better understanding of SLE physiopathology provides to analytic and immunological criteria in the new sets of criteria against the relevance of dermatological manifestations in the ACR-1997 criteria.

Patients who were classified as having SLE by the EULAR/ACR-2019 classification criteria had a mean higher score and were further above the cut-off point than in the other two sets of criteria. We have not found this data described in other cohorts but we consider this finding to be positive, making it more difficult to find doubtful cases. It is worth emphasizing that high scores (≥20) in the EULAR/ACR-2019 criteria were associated with more accumulated damage and this might indicate a predictive role of this set of criteria for disease severity.

We also observed that patients with low SLEDAI scores were less likely to meet the classification criteria for any of the three sets. Thus, 11.4% of patients diagnosed with SLE barely classified in any of the three sets of classification criteria and were characterized by presenting low SLEDAI scores (0.6 ± 0.9). Recently, higher disease activity indices in SLE have been described in patients with increased EULAR/ACR-2019 scores and <12 months of disease course [20]. Other authors conclude that the new criteria may misclassify a small subset of SLE patients with milder disease [17].

Another interesting finding was the different weights that some features have in the different sets of classification criteria. The clinical and immunological characteristics that were statistically associated with positive EULAR/ACR-2019 criteria were renal involvement, arthritis, low C3 and C4 and a positive test for DNA, with all of those important and highly specific to SLE [21]. Those features were also associated with the SLICC-2012 classification criteria together with leukopenia/lymphopenia. None of these two sets showed a statistical association with other clinical domains such as cutaneous, serositis or constitutional syndrome. On the other hand, patients that met the ACR-1997 criteria were linked with other clinical characteristics such as in the cutaneous, serositis, arthritis or hematological domains. It is noteworthy that the magnitude of dermatological manifestations in the ACR1997 classification criteria decreased in favor of analytic and immunological parameters in the later presented classification criteria from 2012 and 2019. Partially, the explanation may lie in the different definitions used for cutaneous lupus among the three sets of criteria and the weight given to every single manifestation [11].

We also observed a low agreement among the different criteria to classify patients as having SLE, with the highest kappa coefficient between the EULAR/ACR-2019 and SLICC-2012 classification criteria (K = 0.61). This may be explained by similarities between the EULAR/ACR-2019 and SLICC-2012 classification criteria [6]. 

Some limitations are obvious in our study. Firstly, this is a retrospective study with the weaknesses of such types of projects. The inclusion criterion that the patients had to accomplish for us to apply every set of criteria excluded patients with previous unavailable data or there was a need to update the information on specific items. Moreover, some clinical information may have been missed or underestimated. This is associated with the second limitation of the study: the small size of the cohort did not allow us to precisely identify clinical profiles of patients diagnosed with SLE who did not meet some or any of the classification criteria. Finally, specificity has not been assessed as we did not include control subjects or patients with other diagnoses. Further research with larger, multiracial and worldwide-representative cohorts is needed, especially regarding the specificity of the EULAR/ACR-2019 criteria. 

## 5. Conclusions

To summarize, our data support the validity of the new 2019 criteria for accurate classification of SLE in patients with longstanding disease, potentially leading to better outcomes and targeted therapies. It also expands the reliability of EULAR/ACR-2019 classification criteria in real-life conditions, in a longstanding disease cohort exposed to several treatments, sometimes with low disease activity.

## Figures and Tables

**Figure 1 jcm-10-02377-f001:**
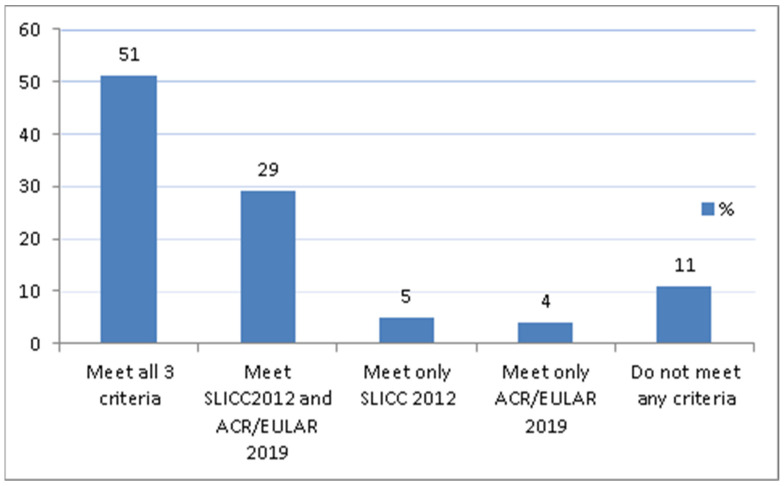
Proportion of patients meeting the different sets of criteria.

**Table 1 jcm-10-02377-t001:** Demographic and clinical characteristics.

**General conditions** **Age (years), mean ± SD** **Female, n (%)** **Disease duration (years), mean ± SD** **SLEDAI score, mean ± SD** **SLICC/ACR DI, mean ± SD**	***n* (79)** **51.89 ± 14.04** **70 (88.6%)** **15.22 ± 11.59** **2.65 ± 2.1** **3.23 ± 1.35**
**Clinical features** Fever, *n* (%) Non-scarring alopecia, *n* (%) Oral ulcers, *n* (%) Subacute cutaneous or discoid lupus, *n* (%) Acute cutaneous lupus, *n* (%) Arthritis, *n* (%) Seizure, *n* (%) Pleural or pericardial effusion, *n* (%) Acute pericarditis, *n* (%) Proteinuria > 0.5 g/24 h, *n* (%) Class II or V lupus nephritis, *n* (%) Class III or IV lupus nephritis, *n* (%) Leukopenia, *n* (%) Thrombocytopenia, *n* (%) Autoimmune hemolysis, *n* (%)	2 (2.5) 18 (22.8) 28 (35.4) 12 (15.2) 16 (20.3) 38 (48.1) 1 (1.3) 2 (2.5) 4 (5.1) 11 (13.9) 15 (19) 12 (15.2) 24 (30.4) 9 (11.4) 2 (2.5)
**Immunologic features** Anticardiolipin IgG > 40 GPL, *n* (%) Anti B2 glycoprotein 1 > 40 UI, *n* (%) Lupus anticoagulant, *n* (%) Low C3 or low C4, *n* (%) Low C3 and low C4, *n* (%) Anti-dsDNA antibody, *n* (%) Anti-Sm antibody, *n* (%)	12 (15.2) 10 (12.7) 12 (15.2) 29 (36.7) 28 (35.4) 50 (63.3) 14 (17.7)

SD: Standard deviation.

**Table 2 jcm-10-02377-t002:** Proportion of patients meeting the different classification criteria and their scores.

	Sensitivity (% Achieving Criteria)	Mean Score of Patients Classified with SLE (±SD)	Mean Score of Patients Not Classified with SLE (±SD)
**ACR-1997**	51.9	5 ± 0.9	2.8 ± 0.8
**SLICC-2012**	87.3	5.3 ± 1.4	2.8 ± 0.4
**EULAR/ACR** **-2019**	86.1	18.6 ± 5.8	6.1 ± 2.5

SD: Standard deviation.

**Table 3 jcm-10-02377-t003:** Statistical associations between the clinical/immunological domains and the EULAR/ACR-2019, SLICC-2012 and ACR-1997 classification criteria.

EULAR/ACR-2019 Domains	Achieving EULAR/ACR-2019 (*p*)	SLICC-2012 Domains	Achieving SLICC-2012 (*p*)	ACR-1997 Domains	Achieving ACR-1997 (*p*)
Fever	0.436	Acute cutaneous lupus	0.265	Malar rash	0.000
Non-scarring alopecia	0.707	Chronic cutaneous lupus	0.130	Discoid rash	0.048
Oral ulcers	0.461	Oral or nasal ulcers	0.697	Photosensitivity	0.000
Subacute cutaneous or discoid lupus	0.770	Non-scarring alopecia	0.820	Oral ulcers	0.034
Acute cutaneous lupus	0.283	Arthritis	0.009	Arthritis	0.000
Arthritis	0.000	Serositis	0.291	Serositis	0.020
Delirium	NA	Renal involvement	0.003	Renal disorder	0.049
Psychosis	NA	Neurologic disorder	0.602	Neurologic disorder	0.957
Seizure	0.583	Hemolytic anemia	0.459	Hematologic disorder	0.005
Pleural or pericardial effusion	0.436	Leukopenia < 4000/mm^3^/lymphopenia < 1000/mm^3^	0.005	Immunologic disorder	0.120
Acute pericarditis	0.267	Thrombocytopenia < 100.000/mm^3^	0.107	ANA	NA
Proteinuria> 0.5 g/24 h	0.059	ANA	NA		
Class II or V lupus nephritis	0.331	Anti-dsDNA	0.022		
Class III or IV lupus nephritis	0.047	Anti-Sm	0.467		
Leukopenia < 4000/mm^3^	0.069	Antiphospholipid antibodies	0.796		
Thrombocytopenia < 100.000/mm^3^	0.471	Low complement	0.022		
Autoimmune hemolysis	0.436	Direct Combs test	0.236		
Anticardiolipin IgG > 40 GPL	0.509				
Anti B2 glycoprotein 1 > 40 UI	0.070				
Lupus anticoagulant	0.270				
Low C3 or low C4	0.980				
Low C3 and low C4	0.031				
Anti-dsDNA antibody	0.000				
Anti-Sm antibody	0.387				

ACR: American College of Rheumatology. EULAR: European League Against Rheumatism. NA: Not applicable. SLICC: Systemic Lupus International Collaborating Clinics.

## Data Availability

All data concerning the results of the study may be found in patient’s clinical charts.

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
