# Peer review of "Assessment of EULAR/ACR-2019, SLICC-2012 and ACR-1997 Classification Criteria in SLE with Longstanding Disease"

_jcm, 2021, doi:10.3390/jcm10112377_

Round 1
Reviewer 1 Report
Congrats to your good paper on the comparison of common SLE classification criteria (ACR/EULAR-2019, SLICC-2012 and ACR-1997). Although the study is retrospective, it provides important results for the validity of these criteria in a real-world setting. The manuscript is basically well-written and structured. I only have minor concerns.
Minor concerns:
In general: some interpunction is missing.
pg 1, line 22: "68 out OF 79"
pg 7, line 203: space is missing after ACR-1997
Author Response
Author's Reply to the Review Report (Reviewer 1)
- Congrats to your good paper on the comparison of common SLE classification criteria (ACR/EULAR-2019, SLICC-2012 and ACR-1997). Although the study is retrospective, it provides important results for the validity of these criteria in a real-world setting. The manuscript is basically well-written and structured. I only have minor concerns.
Thank you very much for these nice comments
- Minor concerns: In general: some interpunction is missing. pg 1, line 22: "68 outOF 79" pg 7, line 203: space is missing after ACR-1997
We have corrected both editing mistakes
Reviewer 2 Report
Major comments:
Materials and Methods:
- Eligibility criteria and methods of selection of patients are not made clear. Were all SLE patients who were actively followed-up included, or did you randomly select a number of patients from your cohort? Were there any exclusion criteria? A full description of the setting and the sampling process is needed in order to assess the generalizability of the results and to ensure authors have tried to overcome selection bias.
-
The authors describe the methods used for measuring immunological tests. Since this was a study including patients with long-standing disease, I assume that immunological testing at baseline had been performed a long time ago, before inclusion in this study and the methods used for testing may have been different at baseline. Did the authors repeat immunological testing in a particular laboratory for all patients before inclusion, in order to ensure a homogeneous method of immunological examination?
- Were there any missing data and if so, how were they addressed?
Results:
- The sensitivities of the three sets of criteria, especially of the ACR 1997 criteria, were found lower than in previous studies, which is rather unexpected, considering that this is a study of patients with long-standing disease. This is not commented in the discussion and a possible explanation should be discussed by the authors.
- The authors mentioned the SLEDAI status in patients not fulfilling any classification criteria. Since this cohort comprised of patients with long-standing disease, the use of a disease activity score at this point does not seem appropriate. Patients with serious SLE manifestations are more likely to have presented with severe disease earlier in their course and may be in remission or LDA at this point. Thus, it would be more appropriate to compare disease severity or damage in patients with long-standing disease.
- Line 127: The phrase "When compared the different domains and the characteristics of the patients that fulfilled the different set of classificatory criteria we found that a 3.7% of the patients as classified only by ACR/EULAR-2019 criteria were characterized by having a positive DNA and hypocomplementemia or arthritis" is not clear. Please restate that phrase.
- It is recommended to check the use of language in the article, mainly in the presentation of results and discussion, since some statements are not clearly phrased.
Minor comments:
- The new classification criteria for SLE have been repeatedly referred to as the "EULAR / ACR criteria", not the "ACR / EULAR criteria", since they were first published (Aringer M, et al. Ann Rheum Dis, 2018). To avoid any misunderstanding, it is suggested to use the name "EULAR / ACR Criteria" in the document.
-
Results:
Table 1: the authors should specify which definition of the different items was used to describe patients' characteristics. For example, the item "leukopenia" or "thrombocytopenia" have different definitions in the ACR 1997, the SLICC 2012 and the EULAR/ACR 2019 classification criteria.
-
Furtherore, were there any ANA negative patients included?
Introduction, line 46: use the verb "missed", instead of "dismissed"
Introduction, line 61: "include", instead of "includes"
Introduction, line 66: "SLICC-2012 criteria sensitivity reached a sensitivity of 95%", omit the first "sensitivity"
Introduction, line 69: Omit the word "the" in the phrase "in 2019 the new classification criteria for the SLE ACR/EULAR came to light"
Author Response
Author's Reply to the Review Report (Reviewer 2)
- Eligibility criteria and methods of selection of patients are not made clear. Were all SLE patients who were actively followed-up included, or did you randomly select a number of patients from your cohort? Were there any exclusion criteria? A full description of the setting and the sampling process is needed in order to assess the generalizability of the results and to ensure authors have tried to overcome selection bias.
Response:
We re-wrote the methods section with more detailed information. “We conducted a retrospective observational study with patients consecutively included during routine clinical follow-up. Our hospital is a tertiary referral center with outpatient specialty clinics of systemic autoimmune diseases. All high complexity and severe diseases are referred to our center through the Universal Health Coverage of the Public Health System. Inclusion criteria were patients older than 18 years, with a confirmed diagnosis of SLE based on the clinical expertise of the attending rheumatologist, with a follow-up of at least 12 months. Exclusion criteria were patients that had lost follow up or that for any reason did not provide necessary clinical or laboratory information to evaluate. A total of 84 patients with SLE were screened during 3 months. Finally, 79 patients were included.
We hope this information gives answer to the reviewer suggestion
- The authors describe the methods used for measuring immunological tests. Since this was a study including patients with long-standing disease, I assume that immunological testing at baseline had been performed a long time ago, before inclusion in this study and the methods used for testing may have been different at baseline. Did the authors repeat immunological testing in a particular laboratory for all patients before inclusion, in order to ensure a homogeneous method of immunological examination?
Response: To ensure the accuracy of the results and to ensure the use of homogeneous methods of immunological exam, all tests were performed in our Immunology Lab and repeated before inclusion and during the follow-up. All patients had blood tests checked at least every 6-12 months except for patients with more active disease that required more frequent examination and blood tests.
- Were there any missing data and if so, how were they addressed?
Response: We tried to ensure the homogeneity of the data and if any patient had missing data or insufficient information to fill all the variables analysed for the classification criteria were excluded (n=5).
- The sensitivities of the three sets of criteria, especially of the ACR 1997 criteria, were found lower than in previous studies, which is rather unexpected, considering that this is a study of patients with long-standing disease. This is not commented in the discussion and a possible explanation should be discussed by the authors.
Response:
We agree with the reviewer and we assume this might be explained by the progressive weight given to analytic and immunological criteria in the new sets of criteria conversely to the former relevance of dermatological manifestations in ACR1997 classification criteria.
- The authors mentioned the SLEDAI status in patients not fulfilling any classification criteria. Since this cohort comprised of patients with long-standing disease, the use of a disease activity score at this point does not seem appropriate. Patients with serious SLE manifestations are more likely to have presented with severe disease earlier in their course and may be in remission or LDA at this point. Thus, it would be more appropriate to compare disease severity or damage in patients with long-standing disease.
Response
Thank you very much for your recommendation. SLICC/ACR Damage Index was analysed and we found interesting results and we have added those in the result section :
- Patients with a higher score (≥20) in EULAR/ACR-2019 classification criteria were characterized by higher SLICC/ACR Damage Index (4.41 ± 1.13) compared to those with lower scores (2.54 ± 1.01 SD) (p<0.02). (Line 146, page 4)
- Patients that did not meet any of the classification criteria presented a significantly lower SLICC/ACR Damage Index compared to those who met all three sets of criteria (0.88 ± 0.56 vs 4.02 ± 0.56; p<0.025). (Line 138, page 4)
- Line 127: The phrase "When compared the different domains and the characteristics of the patients that fulfilled the different set of classificatory criteria we found that a 3.7% of the patients as classified only by ACR/EULAR-2019 criteria were characterized by having a positive DNA and hypocomplementemia or arthritis" is not clear. Please restate that phrase.
Response: We restated the phrase as “7 patients only fulfilled one of the three classification criteria: 3 of them only met EU-LAR/ACR-2019 criteria and were characterized by having hypocomplementemia or arthritis with positive DNA and ANA.”
(Line 141, page 4)
- It is recommended to check the use of language in the article, mainly in the presentation of results and discussion, since some statements are not clearly phrased.
Response: Thank you for the suggestion. We have revised and newly edited the manuscript as suggested.
- The new classification criteria for SLE have been repeatedly referred to as the "EULAR/ACR criteria", not the "ACR / EULAR criteria", since they were first published (Aringer M, et al. Ann Rheum Dis, 2018). To avoid any misunderstanding, it is suggested to use the name "EULAR/ACR Criteria" in the document.
Response: Thank you for the suggestion. Corrections have been made in the revised manuscript.
9.Table 1: the authors should specify which definition of the different items was used to describe patients' characteristics. For example, the item "leukopenia" or "thrombocytopenia" have different definitions in the ACR 1997, the SLICC 2012 and the EULAR/ACR 2019 classification criteria.
Response: We have defined and named the variables in Table 1 as suggested.
10.Furtherore, were there any ANA negative patients included?
Response: All patients had a positive ANA test. We have added this information in the revised manuscript.
- Introduction, line 46: use the verb "missed", instead of "dismissed"
Introduction, line 61: "include", instead of "includes"
Introduction, line 66: "SLICC-2012 criteria sensitivity reached a sensitivity of 95%", omit the first "sensitivity"
Introduction, line 69: Omit the word "the" in the phrase "in 2019 the new classification criteria for the SLE ACR/EULAR came to light"
Response: Thank you for your corrections. They have been made in the revised manuscript.
Reviewer 3 Report
(1) What can the results and conclusions of this study add to previously known ones?
(2) One of the limitations of the 2019 ACR/EULAR classification criteria is excluding ANA-negative SLE patients. I wonder how many ANA-negative SLE patients were included in your cohort.
(3) Do the methods to detect or measure SLE-specific antibodies, which are described in the Methods section, mean those which were used at the time of diagnosis of SLE in all patients in this study? Because the mean disease duration of 15.22 years is not a short period when no advanced methods cannot be developed.
(4) The number of patients included in this study is too small to represent the clinical features of Spanish patients with SLE. If so, this study may provide no additive and valuable information to the previous one. Thus it can be better to include more Spanish patients from other hospitals so that this study may have statistical power.
Author Response
Author's Reply to the Review Report (Reviewer 3)
- What can the results and conclusions of this study add to previously known ones?
Response: To our knowledge, this is the first study to evaluate the performance of the three sets of criteria in a cohort of patients with longstanding disease. There are 5 other studies published in 2020 that evaluated the performance in early SLE, undifferentiated connective tissue disease, paediatric SLE and another one that used a weighted classification of regression. Our data support the validity of the new 2019 criteria endorsing for and accurate classification of SLE in patients with longstanding disease, potentially leading to better outcomes and targeted therapies. It also expands the reliability of EULAR/ACR-2019 classification criteria in real life conditions in a longstanding disease cohort, exposed to several treatments, sometimes with low activity disease.
* Petri M et al. A Comparison of 2019 EULAR/ACR SLE Classification Criteria with Two Sets of Earlier SLE Classification Criteria. Arthritis Care Res (Hoboken). 2020 May 20. doi: 10.1002/acr.24263. Epub ahead of print. PMID: 32433832.
* Adamichou C et al. In an early SLE cohort the ACR-1997, SLICC-2012 and EULAR/ACR-2019 criteria classify non-overlapping groups of patients: use of all three criteria ensures optimal capture for clinical studies while their modification earlier classification and treatment. Ann Rheum Dis. 2020 Feb;79(2):232-241. doi: 10.1136/annrheumdis-2019-216155. Epub 2019 Nov 8. PMID: 31704720.
* Radin M et al. Impact of the new 2019 EULAR/ACR classification criteria for Systemic Lupus Erythematosus in a multicenter cohort study of 133 women with undifferentiated connective tissue disease. Arthritis Care Res (Hoboken). 2020 Jul 23. doi: 10.1002/acr.24391. Epub ahead of print. PMID: 32702197.
*Batu ED et al. The Performances of the ACR 1997, SLICC 2012, and EULAR/ACR 2019 Classification Criteria in Pediatric Systemic Lupus Erythematosus. J Rheumatol. 2020 Nov 15:jrheum.200871. doi: 10.3899/jrheum.200871. Epub ahead of print. PMID: 33191281.
*Rodrigues Fonseca A et al. Comparison among ACR1997, SLICC and the new EULAR/ACR classification criteria in childhood-onset systemic lupus erythematosus. Adv Rheumatol. 2019 May 15;59(1):20. doi: 10.1186/s42358-019-0062-z. PMID: 31092290.
- One of the limitations of the 2019 EULAR/ACR classification criteria is excluding ANA-negative SLE patients. I wonder how many ANA-negative SLE patients were included in your cohort.
Response: All patients were ANA-positive. We added this information in the revised manuscript (Line 129, page 3)
- Do the methods to detect or measure SLE-specific antibodies, which are described in the Methods section, mean those which were used at the time of diagnosis of SLE in all patients in this study? Because the mean disease duration of 15.22 years is not a short period when no advanced methods cannot be developed.
Response: To ensure the accuracy of the results and to ensure the use of homogeneous methods of immunological exam, all tests were performed in our Immunology Lab and repeated before inclusion and during the follow-up. All patients had blood tests checked at least every 6-12 months except for patients with more active disease that required more frequent examination.
- The number of patients included in this study is too small to represent the clinical features of Spanish patients with SLE. If so, this study may provide no additive and valuable information to the previous one. Thus it can be better to include more Spanish patients from other hospitals so that this study may have statistical power.
Response: This was a pilot study from a single center. 84 patients were screened between June 2020 and August 2020 during routine clinical follow-up. Finally, 79 patients were included. The strength of this study is the strict evaluation to ensure data completeness in short period of time in a single center to minimize information bias. We have in mind to collaborate with other Spanish centers to enlarge the sample size in a near future.
Thank you very much for your comments and suggestions. We hope that with all changes made according to your suuggestions the article will now be suitable for publication.
Sincerely
Round 2
Reviewer 2 Report
1) In the manuscript the authors repeatedly refer to "DNA levels", for example, in the Summary, line 27-28, or line 165. It would be appopriate to use the term "anti-double-stranded DNA" or "anti-dsDNA" levels.
2) In scientific writing, it is appropriate to avoid starting sentences with a number. It would be advisable to either spell out a number, for example in line 105 use "Eighty-four patients", or rephrase the sentence, for example "Between June 2020 and August 2020, 84 patients were screened...".
Author Response
1) In the manuscript the authors repeatedly refer to "DNA levels", for example, in the Summary, line 27-28, or line 165. It would be appropriate to use the term "anti-double-stranded DNA" or "anti-dsDNA" levels.
2) In scientific writing, it is appropriate to avoid starting sentences with a number. It would be advisable to either spell out a number, for example in line 105 use "Eighty-four patients", or rephrase the sentence, for example "Between June 2020 and August 2020, 84 patients were screened...".
Response: Thank you very much for the suggestions. Corrections have been made in the revised manuscript. (Page 1, line 26 / page 2, line 56 / page 4, line 151, 164, 166, 167 / Page 5 line 171) and page 3, line 108, 133 /page 4 line 140, 149
We thank you for your comments and suggestions
Reviewer 3 Report
The authors provided appropriate answers to my comments. As the authors insisted, the study investigating the accordance among the three classification criteria for SLE in patients with longstanding SLE seems rare. Therefore, this article has a clinical implication as a pilot study.
However, it may be better to describe the clinical benefit and significance of investigating the accordance among the three classification criteria for SLE in patients with longstanding SLE in real clinical practice in the INTRODUCTION and CONCLUSION sections.
Author Response
1) The authors provided appropriate answers to my comments. As the authors insisted, the study investigating the accordance among the three classification criteria for SLE in patients with longstanding SLE seems rare. Therefore, this article has a clinical implication as a pilot study.
However, it may be better to describe the clinical benefit and significance of investigating the accordance among the three classification criteria for SLE in patients with longstanding SLE in real clinical practice in the INTRODUCTION and CONCLUSION sections.
Response: Thank you very much for your suggestion. We added an emphasis of the clinical benefits and significance of our study in the last two paragraphs of the Introduction and Conclusion sections. (Page 2, line 80-84) and (Page 7 line 260-262)
On behalf of all authors we thank you for your comments and suggestions